# Natural Bioactive Products as Epigenetic Modulators for Treating Neurodegenerative Disorders

**DOI:** 10.3390/ph16020216

**Published:** 2023-01-31

**Authors:** Olaia Martínez-Iglesias, Vinogran Naidoo, Iván Carrera, Lola Corzo, Ramón Cacabelos

**Affiliations:** EuroEspes Biomedical Research Center, International Center of Neuroscience and Genomic Medicine, 15165 Bergondo, Corunna, Spain

**Keywords:** epigenetics, neurodegeneration, epinutraceuticals, DNA methylation, histone modifications, miRNAs

## Abstract

Neurodegenerative disorders (NDDs) are major health issues in Western countries. Despite significant efforts, no effective therapeutics for NDDs exist. Several drugs that target epigenetic mechanisms (epidrugs) have been recently developed for the treatment of NDDs, and several of these are currently being tested in clinical trials. Furthermore, various bioproducts have shown important biological effects for the potential prevention and treatment of these disorders. Here, we review the use of natural products as epidrugs to treat NDDs in order to explore the epigenetic effects and benefits of functional foods and natural bioproducts on neurodegeneration.

## 1. Introduction

Neurodegenerative disorders (NDDs) are major public health concerns in Western countries and are typically associated with aging. The global economic burden of dementia exceeds USD 800 billion and is estimated to cost, on average, USD 30,000–60,000 per person/year [1,2]. Alzheimer’s disease (AD) and Parkinson’s disease (PD) are the most prevalent NDDs globally. AD is degenerative and causes the permanent loss of cognitive function and memory [3,4]. The primary clinical hallmarks of AD are intracellular neurofibrillary tangles that are generated by hyperphosphorylated microtubule-associated tau protein accumulation, as well as senile plaques that are caused by an increase in amyloid-beta (Aβ) load [5]. PD affects 2% of the population over 60 years of age and is the second most common NDD [6]. PD is multifactorial, with environmental, genetic, epigenetic, and cerebrovascular components [7]. The motor dysfunction that is observed in patients with PD is the result of the progressive deterioration of dopaminergic neurons in the substantia nigra *pars compacta*. This decline in neuronal function is accompanied by the formation and accumulation of intracellular inclusions (Lewy bodies) that contain mostly α-synuclein protein [8]. 

Epigenetics is the study of how reversible alterations in gene expression occur without changes to the underlying DNA sequence, linking the genome and the environment [9,10,11]. The accumulation of various epigenetic changes over the lifespan may contribute to cerebrovascular and NDDs [12,13]. DNA methylation, chromatin remodeling/histone modifications, and microRNA (miRNA) regulation are classic epigenetic mechanisms [9,11,13]. DNA methylation involves the transfer of methyl groups from S-adenosyl methionine (SAM) via DNA methyltransferases (DNMTs) to the 5’ position of cytosines in CpG dinucleotides, converting them to 5-methylcytosines (5mC). This mechanism modifies DNA stability and accessibility, thereby controlling gene expression [14]. DNA methylation is commonly a repressive mark [15] and attracts other elements involved in gene-silencing, for example, methyl-CpG-binding proteins [16,17]. The DNMT protein family consists of three members (DNMT1, DNMT2, and DNMT3), all of which are expressed in neurons [18] but have diverse roles. DNMT1 regulates the inheritance of methylation marks and maintains methylation patterns after cell division [19]. DNMT3a- and -3b mediate *de novo* methylation [20,21]. TET1, TET2, and TET3 are methyl cytosine dioxygenases that convert 5mC to 5-hydroxymethylcytosine (5hmC) [22].

Histone acetylation is a reversible process in which an acetyl group is added to a lysine residue at the N-terminus of histones. This epigenetic modification regulates gene expression by promoting gene transcription via the binding of transcription factor and related enzyme complexes to DNA [23]. Histone deacetylases, however, inhibit gene expression [23]. Sirtuins (SIRTs, silent mating-type information regulation 2 homolog in yeast) are nicotine adenine dinucleotide (NAD+)-dependent histone deacetylases (HDACs) that are found in eukaryotes and in bacteria [24]. In humans, the SIRT family consists of SIRTs 1–7. Each of these SIRTs has distinct subcellular localizations, enzymatic activity, and physiological roles, and they are all involved in maintenance of chromatin structure, metabolism, cellular stress response, cell differentiation, cell cycle control, and aging [25]. 

MicroRNAs (miRNAs) are a class of small non-coding RNAs that function in RNA silencing and post-transcriptional regulation of gene expression [26]. In most cases, miRNAs interact with the 3′untranslated region (3′UTR) of target mRNAs to induce mRNA degradation and translational repression [26]. MiRNAs have been proposed as biomarkers for a variety of diseases including cancer and NDDs [27,28]. Various innovative strategies have recently been used to deliver miRNAs into target cells, which suggests that miRNAs may be used as therapeutic agents [28,29].

Medicinal plants have been used since ancient times to treat disease because of their wide range of pharmacological and bioactive properties. Numerous studies have shown that different nutritional compounds and bioproducts have significant modulatory effects on epigenetic apparatus [30,31], for example, in cancer and atherosclerosis. However, our understanding of epigenetic regulation by natural products in brain-related diseases remains limited. The present review highlights the key epigenetic mechanisms through which plant- and animal-derived bioactive compounds act against neurodegenerative processes and the potential of these epinutraceutical bioproducts to counteract NDD pathophysiology. 

## 2. Epigenetics in Neurodegenerative Disorders

Neurophysiological mechanisms such as memory acquisition, learning, and motor coordination are, to a large extent, epigenetically regulated [32,33] (Figure 1). Alterations to the highly regulated epigenetic machinery increase the risk for the onset of various NDDs. These epigenetic aberrations target genes that are linked to synaptic plasticity, immune responses, cell development, and apoptosis [33,34,35,36,37]. 

### 2.1. Role of DNA Methylation in Neurodegenerative Disorders

DNA methylation levels are reduced in the AD and PD brain and in blood samples from animal models and human subjects with NDDs [38,39,40,41]. Downregulated DNMT and impaired vitamin B12 activities are the main factors that contribute to this global hypomethylation. Unlike patients with AD and PD, patients with amyotrophic lateral sclerosis (ALS) show increased DNMT expression and higher levels of DNA methylation than healthy individuals, suggesting that global DNA hypermethylation may be a contributing factor to the disease [42]. 

An increasing amount of data suggest a link between gene-specific methylation and neurodegeneration. A deficiency in vitamin B, for example, decreases glycogen synthase kinase 3β (GSK3β) methylation in AD patients; GSK3β expression is consequently increased, inducing tau phosphorylation, the formation of neurofibrillary tangles (NFTs), loss of cytoskeletal integrity, and cell death [43]. Genes such as bridging factor 1 (*BIN1*), complement receptor 1 (*CR1*), CD33, and tumor necrosis factor (*TNFα*), which are involved in cell death and neuroinflammation, are hypomethylated [44,45]. However, there are also several examples of DNA hypermethylation and decreased expression. Some of these genes include sortilin-related receptor (*SORL1*) and neprilysin (*NEP*), which are both involved in the degradation and clearance of Aβ. Other examples of hypermethylated genes are thromboxane A2 receptor (*TBXA2R*), sorbin, SH3 domain-containing 3 (*SORBS3*), and spectrin beta 4 (*SPTBN4*), which are hypermethylated in animal models of AD and in patients with AD [46]. Thus, blood DNA methylation has been proposed as a biomarker for dementia [38,39,41,47,48].

*SNCA* is a gene that encodes α-synuclein, a protein that is found in many tissues, including the brain. Mutations in the *SNCA* gene have been linked to several neurodegenerative diseases, most notably PD and Lewy body dementia. The promoter in *SNCA* is hypomethylated in blood and brain samples from PD patients, causing an overexpression of α−synuclein and fibrillary aggregation that promotes nigrostriatal degeneration. These patterns of hypomethylation and overexpression are associated with posttranslational modifications of α−synuclein and are also observed in the putamen and cerebral cortex of patients with sporadic PD [49].

### 2.2. Post-Translational Histone Modifications in Neurodegenerative Disorders

Histone modifications such as acetylation or methylation may contribute to the development and progression of NDDs. A correct balance between the activity of HAT (histone acetyltransferases) and that of HDAC (histone deacetylases) is essential for maintaining brain homeostasis [50]. Increased histone acetylation has been implicated in AD pathology, and recent data indicate that HDAC inhibitors are neuroprotective by regulating memory and synaptic dysfunctions in cellular and animal models of AD [51].

Alpha-synuclein binds to histones, prevents H3 acetylation, and induces neurotoxicity [52]. Tri-methylation of histone H3 on lysine 4 (H3K4me3) is increased in the *SNCA* promoter in the substantia nigra in patients with PD. H3K4me3 is a transcription-promoting histone modification that increases gene transcription and expression, suggesting that H3K4 tri-methylation is involved in the induction of α-synuclein overexpression [53]. Long-term treatment with levodopa also causes deacetylation of histone H4 at lysine 5, 8, 12, and 16. Exposure to the neurotoxin 1-methyl-4-phenyl-1,2,3,6-tetrahydropyridine (MPTP) destroys dopaminergic neurons and causes PD-like symptoms, but it also increases the levels of histone H3 acetylation, which are reduced by treatment with levodopa [54]. High levels of histone H2A, H3, and H4 acetylation are present in dopaminergic neurons in post-mortem PD patients [55]. Indeed, treatment with HDAC inhibitors reduces α-synuclein neurotoxicity in neuroblastoma cells and in dopaminergic neurons in the α-synuclein transgenic *Drosophila* model of PD [52,56,57]. The SIRT family of proteins promotes lifespan and healthy aging by modulating a variety of cellular processes, including metabolism, chromatin silencing, cellular differentiation and stress response, inflammation, and cell death [57]. SIRT1, the best-studied member of the SIRT family, is an NAD-dependent protein deacetylase that regulates many important biological processes by removing acetyl groups from target proteins. In recent years, SIRTs have been implicated in neurodegenerative diseases such as AD and PD, and several studies show that SIRTs are neuroprotective [58]. 

Histone methyltransferases (HMTs) and demethylases (HMDs) catalyze the methylation and demethylation of histones, respectively, and have become an innovative target for treating or preventing NDDs. A dynamic balance between HAT and HMT regulates suppressive chromatin, which is associated with NDD pathology and progression [59]. In pre-plaque AD transgenic mice, H3K14 and H3K9me2 histone methylation levels are elevated [60], whereas during the progression of PD, the demethylase Jumonji domain-containing protein-3 (Jmjd3) is essential for modulating microglia phenotypes [61].

### 2.3. Regulation of micro-RNAs in Neurodegenerative Disorders

MicroRNAs (miRNAs) are a class of short noncoding RNA molecules that regulate the expression of genes involved in several cellular processes such as differentiation, proliferation, and cell death. They have been implicated in several neurodegenerative diseases, including AD, PD, and Huntington’s disease (HD), in which they are involved in neuroinflammation and cell death [62]. MiRNAs can be detected and quantified in peripheral biofluids, such as plasma, serum, and cerebrospinal fluid, and in peripheral blood mononuclear cells (PBMCs), suggesting that circulating miRNAs extracted from blood or other biofluids may serve as non-invasive and cost-effective biomarkers for the early detection of NDDs such as AD and PD. MiRNAs could therefore be utilized as a screening tool for the early detection and monitoring of NDD progression. In AD, miRNAs regulate synaptic activity, and several miRNAs (e.g., miR-124, miR-125b, miR-34c, and miR-132) are enriched in affected synapses [63,64]. Dysregulation of brain-specific miRNAs adversely modulates synaptic activity in AD by suppressing their target genes, thus impairing synaptic activation and transmission. This has pathogenic consequences such as neurotrophic and synaptic deficits and astrogliosis. Siedlecki-Wullich et al. (2019) showed that a plasma miRNA signature comprising miR-210-3p, miR-181c-5p, and miR-92a-3p may be effective as a non-invasive clinical biomarker for the diagnosis of AD, and which could be utilized to improve future treatment strategies [65]. MiRNAs may, therefore, be informative and specific biomarkers for evaluating the staging, progression, and prognosis of NDDs [45].

## 3. Treatment of Neurodegenerative Disorders with Epinutraceutical Bioproducts

Pharmacogenetic studies using conventional pharmaceuticals against NDDs show that epigenetic regulation is important for the expression of pathogenic, mechanistic, metabolic, transporter, and pleiotropic genes involved in the pharmacogenetic network responsible for NDD treatment efficacy and safety. Given that many epigenetic alterations are reversible, it is likely that nutraceutical products, specific bioproducts with epigenetic effects, and novel epigenetic drugs with no side effects and the ability to access the brain will, in the future, provide some benefit in terms of prevention and early treatment of patients with NDDs. There has been a surge in the number of clinical trials examining the use of epidrugs for treating disease. Epidrugs are pharmacological compounds that selectively target enzymes with epigenetic activity or the epigenome. Despite a few encouraging findings, several of these drugs have been linked to considerable adverse effects [66]. Nutraceuticals, however, have various benefits over conventional medications, including a lower incidence of adverse/side effects, cost-effectiveness, widespread availability, diverse therapeutic effects, and the ability to enhance overall patient health [67]. Together, these attributes make nutraceuticals a useful alternative to conventional medications for treating a variety of medical disorders. Dietary supplements have a wide range of biological effects, some of which occur through epigenetically regulated gene expression. Recent advances have shown the epigenetic effects of natural compounds against NDDs [68]. Polyphenols (e.g., resveratrol), isoflavones (e.g., genistein), isothiocyanates (e.g., sulforaphane), γ- and α-tocopherols, and a number of vitamins (e.g., A, C, E, B6, B9, and B12) offer positive health benefits, and several regulate DNA methylation, histone modifications, and miRNA expression [69] (Table 1).

### 3.1. Vitamins

Dietary-derived compounds are generally considered safe. However, because of the complex chemical composition of several natural compounds, data on their toxicity profiles are limited. Therefore, it cannot be assumed that these substances are biologically safe and that they are devoid of any potential health risk. In fact, certain food components exhibit beneficial effects at physiological doses but may be detrimental at pharmaceutical doses. 

Fat-soluble vitamins tend to accumulate in the body, making them more toxic than water-soluble vitamins. High doses of antioxidant vitamins (e.g., vitamins C, E, and/or β-carotene) may have negative consequences on cardiovascular function [70]. Because vitamin A is fat-soluble and may be stored in the liver, consuming high doses of it over time can cause skin, bone, and liver toxicity. Although carotene is converted to vitamin A in the body, when taken as a supplement over extended periods of time, excessive consumption of β-carotene has been linked to an elevated risk of lung cancer [71]. Vitamin C has antioxidant properties, but under conditions of high oxidative stress, it may act as a pro-oxidant (Alissa and Ferns, 2012). Furthermore, excessive vitamin C supplementation causes diarrhea, hyperoxaluria, and hemolysis in individuals deficient in glucose-6-phosphate dehydrogenase, and iron overload in those with thalassemia [72,73]. Hypercalcemia is the most prominent effect of the excessive consumption (a single large dose or excessive pharmacologic dosages) of vitamin D over an extended period. Here, toxicity may be caused by increased blood levels of 25-hydroxycholecalciferol (25-OH-D) and calcium [74]. Long-term treatment with vitamin E increases the risk of heart failure and death, particularly in individuals with chronic illnesses [75]. The primary disadvantage of high vitamin E levels is enhanced, defective blood coagulation caused by vitamin K deficiency due to malabsorption or treatment with anticoagulants [76]. Moreover, tocopherol also has a pro-oxidant effect that is believed to contribute to the increase in fatal myocardial infarctions in clinical trials following supplementation with vitamin E [76]. 

The effects of vitamin B therapy on the brain are significant, and a deficiency in vitamin B12 has been linked to neurologic problems, poor cognition, and AD [77,78]. Treatment with vitamin B12 restores DNA methylation patterns and is an effective supplement in the pharmaceutical cocktail used to treat NDDs [79,80]. Demethylation of the presenilin1 (PSEN1) promoter occurs in human neuroblastoma cells and in transgenic mice maintained under vitamin B-deficient experimental conditions. PSEN1 hypomethylation increases PSEN1, BACE1, and APP protein levels and induces Aβ deposition in the mouse brain [81,82]. In rodent models of AD and in patients with AD, supplementing the diet with folate and vitamins B6 and B12 reduces tau hyperphosphorylation and accumulation in the hippocampus and cortex and ameliorates memory deficits [83,84]. A deficiency in vitamin B12 affects DNMT1, DNMT3a, and DNMT3b activity [85,86]. Moreover, vitamin B deficiency during pregnancy alters AD-implicated genes and reduces global methylation levels in the offspring [87]. Moreover, vitamin B6 decreases the risk of PD [88].

After ingestion, folate (vitamin B9) is converted to tetrahydrofolate, which is involved in the remethylation of homocysteine to methionine [31], and SAM is converted to S-adenosyl-L-homocysteine (SAH), which is an inhibitor of methylation reactions (Figure 2). Supplementing the diet of mothers with folate during the early stages of pregnancy protects against neural tube defects [89]. Epidemiological and experimental data link folate deficiency and resulting low homocysteine levels to several neurodegenerative conditions including stroke, AD, and PD [90]. Folate intake is associated with a reduced risk of AD [91]. Furthermore, serum folate deficiency is linked to a greater incidence of dementia [92]. In patients with AD, serum homocysteine (Hcy) levels are substantially higher, and serum folate and vitamin B12 levels are lower than in non-dementia patients [93]. In patients with dementia, supplemental treatment with folic acid (1.25 mg/d for six months) causes a slight increase in cognitive function, as assessed with the mini-mental state examination (MMSE) [94]. Folic acid treatment increases SAM and SAM/SAH levels, and Aβ40, PS1 mRNA, and TNFα mRNA levels are low in AD patients treated with folic acid (ClinicalTrials.gov Identifier: ChiCTR-TRC-13003246) [94]. Folate supplementation also improves cognitive function by decreasing the levels of peripheral inflammatory cytokines [95]. A recent systematic review and meta-analysis shows that plasma/serum folate levels are lower in AD patients than in control subjects [96]. Plasma folate and vitamin B12 levels are both low in PD patients [96], and patients treated with vitamin B12 and folate show a significant decrease in Hcy levels [97]. A high dietary intake of vitamin B6 is associated with a considerably reduced risk of PD [98]. Higher levels of vitamin B12 at PD diagnosis are associated with a decreased risk for the future development of dementia [98]. Moreover, low vitamin B12 levels have been associated with peripheral neuropathy, cognitive impairment, and faster progression of PD [99].

Vitamins A, C, and E have strong antioxidant properties. Oxidative damage causes DNA to become a poor substrate for DNMTs, thereby producing a state of global hypomethylation [78,100]. Oxidation may also change the histone acetylation balance. As a result, the antioxidant capabilities of those vitamins may help to decrease aberrations in acetylation status [78]. The metabolically active derivative of vitamin A, retinoic acid, alters the levels of SAM and SAH [101]. A retinoic acid-deficient diet reduces the levels of DNA methylation by altering the availability of methyl groups [102], and treatment with Vitamin A has been shown to improve cognitive functions such as memory [77]. Several NDDs are associated with oxidative stress; vitamin E acts as an antioxidant, minimizing NDD pathogenesis [103,104]. Vitamin C is a critical antioxidant molecule in the brain. Under physiological pH conditions, vitamin C is mostly present as an ascorbate anion. Ascorbate is a cofactor for methylcytosine dioxygenases, which are responsible for DNA demethylation, as well as a possible cofactor for certain Jumonji C domain-containing histone demethylases, which catalyze histone demethylation [105]. Vitamin C deficiency may play a role in neurocognitive dysfunction and impairment, and there is evidence indicating that ascorbic acid is therapeutically beneficial against aging-related diseases including NDDs [105].

### 3.2. Curcumin

Curcumin, a phenolic compound, is the major curcuminoid in the herb *Curcuma longa* (turmeric). In addition to the wide range of therapeutic effects of curcumin, there are concerns regarding its potential for toxicity and its adverse effects. For example, curcumin inhibits glutathione S-transferase (GST), impairing detoxification and thereby causing potentially adverse drug–drug interactions. Curcumin also inhibits the human ether-a-go-go-related gene (hERG) channel, inducing cardiotoxicity. Furthermore, high concentrations (>10 µM) of curcumin are cytotoxic to human gastric adenocarcinoma cell lines [106]; curcumin (0.4 mg/mL) is also strongly cytotoxic to lymphocytes from normal donors and from patients with leukemia [72]. The US Food and Drug Administration currently considers turmeric to be generally recognized as safe (GRAS) with low toxicity [107].

Curcumin is not genotoxic or mutagenic [108]. In mice, following oral dosing of curcumin, the LD50 was over 2000 mg/kg; single oral doses (1–5 g/kg body weight) of curcumin did not produce toxicity in rats [109]. The highest recommended daily dose of curcumin ranges from 3 mg/kg to 10 g [110]. However, in a clinical trial, there were no major side effects in healthy male and female volunteers who took a daily dosage (12 g, capsule) of a curcuminoid formulation (curcumin, bisdemethoxycurcumin, and demethoxycurcumin) [111]. Moreover, a phase I clinical trial showed that long-term oral intake of curcumin (8 g of curcumin/day/three months) is not toxic in humans. Other clinical studies further showed that oral doses of curcumin (1.1–2.5 g/day; and 6 g/day for 4–7 weeks), are safe [112]. Curcumin (500–8000 mg/day/3 months) taken by individuals with cardiovascular risk factors has a favorable safety profile. This profile is comparable in individuals with pancreatic (8 g curcumin/day for 2 months) [113], breast (6 g mg curcumin/day), and colorectal cancer (36–180 mg curcumin/day for 4 months) [114,115]. 

Pharmacokinetic data show that curcumin has low water solubility (0.6 µg/mL) [116], which indicates its low bioavailability [116]. The hydrophobicity of curcumin reduces its absorption from the digestive tract, and a significant part of the curcumin molecule is inactivated via metabolism in the liver. Its plasma concentration does not exceed 1 µmol, and its peak concentration is detected 1–2 h after ingestion [117]. The maximum plasma concentrations of curcumin are 0.05 µg/mL (from 12 g, p.o.) in humans, 1.35 µg/mL (2 g/kg) in rats, and 0.22 µg/mL (1 g/kg) in mice [118]. Most curcumin after oral intake is eliminated in the feces. More specifically, following the administration of curcumin (1 g/kg body weight, p.o.) to rats, approximately 75% of curcumin is eliminated in the feces [116]. Curcumin is sensitive to pH-dependent degradation; under alkaline conditions, curcumin is degraded within 30 min, whereas under acidic conditions, curcumin degradation is significantly slower (<20% of total curcumin) and occurs within 60 min [119]. Curcumin is also rapidly transformed into water-soluble metabolites (sulfates and glucuronides) through hepatic first-pass metabolism and excreted in urine.

New formulations of curcumin that are being developed for improved bioavailability rely on plasma levels of glucuronidated curcumin to assess quality. However, it is unclear whether this accurately reflects brain tissue levels, particularly because high levels of the parent compound accumulating in the brain can occur in the absence of detectable levels of curcumin in the blood [119]. One formulation, Meriva, involves complexing curcumin with phosphatidylcholine, which protects curcumin from degradation while also increasing its uptake across lipophilic cell membranes [120]. In a randomized, double-blind study in humans, curcumin absorption and bioavailability with Meriva were substantially higher than with unformulated curcumin. Curcumin has also been incorporated into a solid-lipid-particle-based formulation termed LongVida^®^ [121]. This formulation prevents curcumin from rapidly degrading and has undergone extensive safety testing in rodents; the no-observed-adverse-effect-level (NOAEL) for LongVida^®^ was 720 mg/kg body weight/day, for a period of 90 days [122]. Compared to unformulated curcumin, LongVida^®^ increases (systemic) curcumin half-life and improves its bioavailability in healthy volunteers and in patients with late-stage osteosarcoma. Piperine, an inhibitor of the P-glycoprotein drug transporter, has been used in formulations for improving the oral absorption of curcumin [123] and improves the bioavailability of curcumin by approximately 2000%. Nanoparticle-based formulations of curcumin have been extensively investigated to improve the absorption of curcumin and to prevent adverse drug reactions. One such formulation is Theracurmin, a colloidal curcumin dispersion (glycerin and ghatti gum) which, in healthy patients, has increased absorption and clinical bioavailability that is at least 16-times higher than that of unformulated curcumin [124]. Various studies have used either whole turmeric extract, different lipidated formulae of curcumin, or nano-curcumin, and these data need to be compared directly to the effects of natural curcumin. Furthermore, the purity of curcumin, as well as a detailed description of the extraction procedure and sources, are crucial elements to consider when determining the efficacy of therapies that use curcumin. In addition, curcuminoid contains two additional compounds (bisdemethoxycurcumin and demethoxycurcumin), yet studies frequently use “curcuminoid” and “curcumin” interchangeably. Therefore, when evaluating the findings of such studies, this clarification must be taken into consideration. Next, it is important to consider that the pharmacokinetics and pharmacodynamics of curcumin vary depending on the route of administration, which impacts therapeutic efficacy. Finally, the duration of treatment with curcumin can affect its efficacy. Short- and long-term therapy with curcumin show mixed effects. Although numerous studies have reported the short-term beneficial effects of curcumin, there may an under-reporting of toxic effects in longer-term studies, which emphasizes the need for further work that tests safety and toxicity in relation to the chronic administration of curcumin. There is a general consensus, however, that curcumin has enormous potential as an effective nutraceutical due to its low toxicity, low cost, and ease of availability [125].

Curcumin induces alterations such as changes in pharmacokinetic parameters (peak plasma concentration—C_max_; and area under the curve—AUC) when combined with cardiovascular medications, antidepressants, anticoagulants, antibiotics, chemotherapeutic agents, and antihistamines. The mechanisms that underlie these interactions include inhibition of cytochrome (CYP) isoenzymes and P-glycoprotein. Only a single clinical trial has shown a significant alteration in the function of conventional drugs when used in conjunction with curcumin, therefore highlighting the necessity for further human trials [126].

Curcumin has strong anti-inflammatory, anti-oxidant, and anticancer properties and is a potent regulator of the epigenome [127]. This epigenetic regulation includes inhibition of DNMT expression, modulating methylation, acetylation, ubiquitination, and phosphorylation of histones [127,128] and the regulation of miRNAs [127] (Figure 3). It is unclear whether curcumin decreases DNA methylation by binding to DNMT1 and inhibiting the catalytic thiolate of DNMT1, as other authors do not observe the pharmacological effect of curcumin to be that of a DNMT inhibitor [128].

The ability of curcumin to cross the blood–brain barrier (BBB) is important to its development as a neuroprotective agent. After oral ingestion, curcumin can be detected in the rat brain over a 96 h period, with a peak effect observed at 48 h [129]. Curcumin administered to mice via oral gavage (0.4 µmol), intraperitoneal (0.4 µmol), and intramuscular injections (0.2 µmol) has also been detected in the mouse brain [130]. A detailed assessment of several studies evaluating the ability of curcumin to cross the BBB showed that although curcumin accumulates in the brain, its bioavailability is limited and is short-lived [131]. Therefore, increasing the accumulation of curcumin in the brain is a very important parameter that is necessary for evaluating the neurotherapeutic potential of this compound.

Curcumin is used to treat dyspepsia, stress, and mood disorders [132] and is a specific inhibitor of p300/CREB-binding protein (CBP) HAT activity [133]. In some brain diseases, HAT inhibition decreases nuclear histone acetylation, leading to inhibition of the NF-κB pathway [100,102,103]. P300/CBP acetylates NF-κB, and curcumin inhibits p-300 mediated acetylation of the isoform of NF-κB RelA [134]. Blocking p-300 HAT activity causes pro-nociceptive (BDNF and Cox-2) acetylation in neuropathic pain [135] and modulates DNA damage pathways [136]. Curcumin is also a pan-HDAC inhibitor and may regulate neural-stem-cell-directed neurogenesis [137]. Curcumin is a highly potent HDAC inhibitor and is more efficacious than the well-characterized HDAC inhibitors valproic acid and sodium butyrate [78]. Curcumin decreases HDAC-1, -3 and -8 expression, which increases histone-4 acetylation [131,138], and enhances miRNA-22, miRNA-186a, and miRNA-199a levels [139].

Curcumin treatment inhibits HAT and HDAC activity in AD [100]. HDAC alterations to histone acetylation at the promoter regions of AD-related genes such as *PSEN1* and beta-site amyloid precursor protein cleaving enzyme 1 (*BACE1*) were studied in the mouse neuroblastoma N2a cell line. Curcumin inhibits p300 and significantly suppresses PS1 and BACE1 expression by inhibiting H3 acetylation at their promoter regions [140]. The combinatorial effect of SAHA and curcumin is protective against neurotoxic Aβ25-35-induced neuronal damage in pheochromocytoma-derived (PC12) cells via HDAC regulation [141]. Curcumin (Theracurmin, 90–180 mg/day, for 18 months) also possesses p300/CBP HAT inhibitory function and is currently in phase II clinical trials to determine its capability for improving cognitive function and for its Aβ-reducing potential in patients with AD (ClinicalTrials.gov Identifier: NCT01383161) [137,142,143]. In another clinical trial, the effects of a mixture of the three curcuminoids (Super Curcumin C-3 Complex, 1–4 mg/day, for 16 weeks) on the interaction between epigenetic (histone modifications) and inflammatory (inducible nitric oxide synthetase) signaling were investigated in patients with schizophrenia or depression (ClinicalTrials.gov Identifier: NCT 01875822); the outcome of this trial is pending. Curcumin induces the expression of miRNA-22, miRNA-186a, and miRNA-199a in individuals without dementia [144]. These miRNAs, however, are reduced in subjects with AD [140].

### 3.3. Resveratrol

Resveratrol is a polyphenol compound found in a variety of plants, including vegetables, berries, grains, roots, seeds, tea, and grapes. It has anti-inflammatory properties and is a potent antioxidant with positive effects against cancer and brain-related diseases [145]. Resveratrol intake in humans ranges from 700 to 1000 mg/kg body weight/day without causing toxicity [146]. Moreover, the consumption of resveratrol up to 2 g/day is safe if taken in a short period of time [147], and a number of clinical trials show that resveratrol is safe even up to 5 g/day [148]. Resveratrol has an excellent tolerance and safety profile, and there is no known clinically significant pharmacological interaction between this compound and conventional medications [149]. However, recent in vivo studies and clinical trials suggest the possibility of drug–drug interactions at high dosages of resveratrol [150]. At high concentrations, carcinogenic or genotoxic effects of polyphenols are possible because these compounds tend to become pro-oxidants by increasing proliferative signaling via reactive oxygen species (ROS) generation, by disrupting second messenger signaling, and by promoting tumor growth [151].

The use of resveratrol as a pharmaceutical agent is limited by its rapid metabolism and its low bioavailability. More than 70% of orally ingested resveratrol is absorbed in the gastrointestinal (GI) tract but is then processed by three separate metabolic pathways, resulting in its very low bioavailability. Furthermore, resveratrol has a low water solubility (<0.05 mg/mL), which, together with temperature and pH, limits its absorption [152]. Resveratrol reaches its maximum concentration in plasma 30–90 min after oral intake. Peak plasma concentrations of resveratrol are 1–5 ng/mL (4–20 nM) after a single oral dose of 25 mg resveratrol and 2.3 mM after a dose of 5 g resveratrol [153]. In the body, resveratrol forms complexes with transporters such as low-density lipoprotein, albumin, and integrins or is subjected to passive diffusion [154]. Resveratrol is hydrolyzed to oligomeric phenolics and/or undergoes isomeric conversion in the stomach. Furthermore, resveratrol is glycosylated by gut bacteria and may also be modified by hepatic and intestinal conjugation processes. In the liver, resveratrol passes through phase II metabolism, creating methylated, sulfated, and glucuronidated metabolites [155]. Interestingly, resveratrol enhances its own metabolism by increasing the activity of its metabolites and phase II liver detoxifying enzymes [156]. Resveratrol may form complexes with lipoproteins and human serum albumin, which promote resveratrol activity and stability because plasma transport proteins are natural reservoirs of resveratrol in vivo. In humans, resveratrol metabolites are found in high concentrations in urine and plasma, with a half-life that is 10-times longer than native resveratrol [157]. 

Although resveratrol has relatively poor systemic bioavailability, accumulation of resveratrol in epithelial cells throughout the aerodigestive tract (and possibly active resveratrol metabolites) may nevertheless have cancer-preventive and other benefits [158]. Because most polyphenols cross the BBB, they are commonly used in the treatment of many neurodegenerative diseases. Several studies show that resveratrol has neuroprotective effects in animal models of AD and PD [137,144], but its therapeutic applicability is restricted due to its rapid metabolism and low bioavailability. Structural modifications in the resveratrol molecule, such as glycosylation, alkylation, halogenation, hydroxylation, methylation, and prenylation, may aid the synthesis of derivatives of resveratrol that have increased bioavailability and pharmacological activity [152]. Despite this, resveratrol is a promising compound from which useful and more effective derivatives may be created through chemical interventions, for example, glycosylation of resveratrol into the stilbenoid glucoside piceid [152]. In vitro, piceid shows greater scavenging efficacy against hydroxyl radicals than resveratrol [158]. Consequently, the synthesis of resveratrol analogues with improved bioavailability and solubility may help increase the number of targets that are affected by biological molecules by better delineating their biochemical pathways/mechanisms of action. This would open new avenues for the discovery and development of novel agents to treat neurodegenerative diseases.

Resveratrol and its derivatives act as SIRT activators in animal models of neurodegeneration (Figure 4). In mixed primary neuron rat cortical cultures, resveratrol is protective against Aβ toxicity through SIRT1 activation and the inhibition of NF-κB signaling [159]. This neuroprotective property of resveratrol may be mediated in part by the SIRT1-serine/threonine protein kinase ROCK1 [160] and SIRT1-Akt pathways [161], which enhance α-secretase activity. In response to treatment with resveratrol, mouse models of AD show reduced hippocampal damage, a decrease in Aβ plaque formation, and improved memory [162,163]. Compared to preclinical data, there are fewer studies on the effect of resveratrol on cognitive function in AD patients. However, a recent systematic review shows that resveratrol delays cognitive impairment in patients with AD [161]. *p53* is a substrate of SIRT1, and resveratrol allosterically modulates SIRT1, which deacetylates p53, reducing its activity. This inhibition of p53 may disturb the interaction between p53 and GSK3β, both of which are involved in the apoptotic pathway; in AD, p53 activity promotes tau phosphorylation, and GSK3β overactivation increases the amount of neurofibrillary tangles and plaques [164]. Resveratrol also reduces miR-124 and miR-134 expression, promoting brain-derived neurotrophic factor (BDNF) synthesis in mice [165,166]. Moreover, polyphenols revert the modulatory effect of miRNAs in APOE transgenic mice [80]. Resveratrol reduces the expression of miR-15, which is implicated in tau hyperphosphorylation [167], and increases the levels of miR-21, miR-155, miR-125b, and miR-146a, which are related to neuroinflammation [167]. Resveratrol also reduces BACE1 activity by decreasing miR-29c, miR-186, miR-107, miR-9, miR-29a, and miR-188-3p expression and inhibits Aβ accumulation by reducing the expression of miR-124, miR-17, miR-101, miR-16a, and miR-1066 [167]. Therefore, SIRT-activating compounds are potential candidates for treating AD.

For over two decades, mitochondrial dysfunction has been linked to PD etiology. In cultured primary fibroblasts from patients with familial PD, which is linked to different *Park2* mutations, resveratrol improves mitochondrial respiratory capacity through a mechanism involving SIRT1-AMP-activated protein kinase (AMPK) and peroxisome proliferator-activated receptor-γ coactivator (PGC)-1; specifically, resveratrol activates the AMPK-SIRT1 signaling pathway, inducing PGC-1 activity [168]. The effect of PGC-1 activation on mitochondrial respiration enhances mitochondrial biogenesis and increases mitochondrial oxidative function, which represent potential targets for PD therapy. In addition, resveratrol-induced SIRT1 activation and expression are protective against α-synuclein aggregation. Here, SIRT1 deacetylates and activates heat shock factor 1, impacting the transcription of heat shock proteins (Hsp70s), which in turn prevent the production of pathological protein aggregates [169]. Resveratrol is also a histone modifier by inhibiting the enzymes histone acetyltransferase and HDAC [170]. In a human-derived neuroblastoma (SH-SY5Y) cell line, resveratrol reduces rotenone-induced cell injury and *p53* expression; mechanistically, SIRT1 targets histone H3K9 and inhibits *p53* gene expression [171]. Treatment with resveratrol improves the PD phenotype by suppressing neuronal death via modulation of the MALAT1/miR-129/SNCA signaling pathway [172]. Indeed, miRNA-214 contributes to the neuroprotective effects of resveratrol by inhibiting α-synuclein expression in MPTP-treated mice [173].

### 3.4. Epigallocatechin-3-Gallate (EGCG)

Epigallocatechin gallate (EGCG) is the main polyphenolic compound in green tea. Green tea catechins have been linked to the benefits and harmful effects of green tea [174,175]. The principal safety concern arises from case studies that claim a positive correlation between the use of concentrated green tea extract and liver damage [176]. Various factors such as general liver health, individual idiosyncrasies, and genetic factors may contribute to the potential hepatotoxicity associated with consuming green tea. Data from human subjects are inconsistent, with instances of hepatotoxicity reported with various formulations, a range of doses, and various treatment durations. Furthermore, determining the level of EGCG (and green tea extract) in supplements is challenging. Yates et al. (2017) proposed tolerated upper intake limits (TUL) of orally ingested 322 mg EGCG/day (a 70 kg adult) and 300 mg EGCG/day in healthy people based on animal toxicity and human intervention/case/adverse event data, respectively; the recommended acceptable daily intake (ADI) is 4.6 mg EGCG/kg/day [177]. Dosages over 800 mg of EGCG per day (≥4 months) caused increases in aspartate transferase and alanine transaminase levels in fewer than 10% of the population [178]. A comprehensive assessment of published toxicological and human intervention studies to assess the risk to human health from green tea consumption revealed that excessive levels of catechins cause hepatotoxicity in a dose-dependent manner, and that an observed safe level of 704 mg EGCG/day is recommended for human consumption when EGCG is ingested on a regular basis (e.g., in tea) [179]. Overall, there is no recent evidence that invalidates the conclusion by the European Food Safety Authority that a daily intake of 800 mg of EGCG is safe. Moreover, a NOAEL for EGCG has not been scientifically established. The stability of EGCG is a concern, as it is prone to decomposition through epimerization or auto-oxidation by light, temperature, changes in pH, or the presence of oxygen [180,181]. A multicenter, prospective, double-blind, placebo-controlled, randomized pilot study is currently investigating the safety and tolerance of EGCG in boys with Duchenne muscular dystrophy (ClinicalTrials.gov identifier: NCT01183767). 

The bioavailability of EGCG following oral administration is relatively low, which poses a significant limitation for its therapeutic use [181]. Pharmacokinetic data in dogs and rodents show that EGCG is rapidly absorbed in the GI tract, distributed, and metabolized in the liver and colon; EGCG can be reabsorbed from the gut via enterohepatic recirculation. The resultant metabolites are eliminated by the urinary and biliary systems. Following a single oral dose (2 mg/kg body weight in human subjects), traces of EGCG are detectable in urine [182]. Fasting, fish oil, piperine, vitamin C, and albumin improve EGCG bioavailability, whereas oxidation, glucuronidation, sulfation, catechol-O-methyltransferase polymorphisms, calcium, magnesium, and exposure to metals decrease EGCG bioavailability [183]. Structurally, the galloyl moiety in EGCG is responsible for its strong inhibitory activity, and the hydroxyl groups on carbons 3′, 4′, and 5′ of the EGCG molecule possibly contribute to its very high antioxidant activity [184].

In animal studies, EGCGs are poorly absorbed after oral administration; the absolute bioavailability of EGCG in rats and mice is only 1.6% and 26.5%, respectively. This may be because of first-pass effects, which are caused by extraction in the liver and/or via GI metabolism shortly after absorption. Following a single oral dose (250 mg/kg body weight) to dogs, the absorption of EGCG was rapid and reached a maximum concentration in plasma approximately 1 h later [185], which is similar to that in humans [186]. In healthy participants, 26 h after a single oral dose of EGCG (1 g), the peak plasma concentrations of EGCG exceeded 1 µM [187]. In those patients, the relative bioavailability of EGCG was 1.6% (75 mg/kg, body weight) and 13.9% (250 mg/kg, and 400 mg/kg body weight), the C_max_ was 3392 ng/mL, the time to reach C_max_ (T_max_) was 60–115 min, and the AUC (0–∞) was 442–10,368 ng h/mL; the elimination half-life was 5–6 h.

At low or moderate concentrations (plasma levels ≤ 10 µM), EGCG induces antioxidant activity mainly via the production of low quantities of ROS required for the promotion of cellular protection [188]. However, at high concentrations (plasma levels > 10 µM), EGCG exhibits primarily pro-oxidant activity by increasing autophagy and causing cell death [189]. EGCG binds exclusively to the 67-kDa laminin receptor (67LR), which is likely the main cell surface receptor responsible for its anti-tumoral action. This interaction triggers the apoptotic signaling cascade Akt/eNOS/NO/cGMP/PKCδ and causes an increase in cyclic guanosine monophosphate (cGMP) [190]. Furthermore, EGCG inhibits TLR4 signaling via the 67LR-dependent pathway, reducing inflammation [191]. EGCG also alters cell surface growth factor receptors, particularly receptor tyrosine kinases, which are involved in angiogenesis, cell survival, and proliferation [192]. 

The multifaceted actions of EGCG have been extensively studied, with particular emphasis on its potential role in cancer prevention and treatment. However, recent studies have also highlighted the potential therapeutic benefits of EGCG across a wide range of other disorders, including AD, PD, respiratory-, cardiovascular-, and metabolic diseases (e.g., diabetes mellitus and obesity). The neuroprotective properties of EGCG and its metabolites have been extensively studied, and there is a positive correlation between frequent tea consumption and enhanced cognitive function or the mitigation of cognitive impairment [193,194]. Intragastric administration of EGCG prevents cognitive deterioration in senescence-accelerated mice and decreases the accumulation of β-amyloid. EGCG, furthermore, attenuates β-amyloid-induced cognitive impairment by regulating secretase activity through ERK and NF-κB [195]. EGCG also prevents fibrilization of misfolded proteins and restores respiration levels and membrane potential in mitochondria isolated from the hippocampus, cortex, and striatum [35]. Moreover, EGCG, possibly by reducing α-synuclein aggregation [196], is therapeutically beneficial in PD [197], indicating that regular tea consumption may protect against this disorder. However, despite numerous studies on the neuroprotective properties of EGCG, data on its bioavailability in the brain remain limited, which is a critical consideration for the potential therapeutic benefits of EGCG [182]. For EGCG to be recommended as a targeted prevention and individualized treatment for patients, there is a need for further research to be conducted on: its bioavailability in plasma and in the brain, dose–response effects, safety, tolerability, efficacy, and possible interactions with other drugs in a disease-specific manner [182].

The scientific literature on the epigenetic effects of EGCG in neurodegenerative diseases, however, is relatively sparse. Nonetheless, it is clear that EGCG is a DNMT inhibitor that effectively alters the DNA methylation profile [139]. EGCG is currently in phase II and III clinical trials to test its effects on the prevention of Aβ aggregation as a precursor of toxic oligomers in AD (ClinicalTrials.gov Identifier: NTC00951834) [198]. Furthermore, EGCG acts as an HDAC inhibitor and increases NEP expression, facilitates the degradation of Aβ, and attenuates cognitive deterioration in the senescence-accelerated mouse-prone 8 (SAMP8) mouse model of AD [199]. The epigenetic mechanism through which this occurs in brain diseases is unclear. However, evidence, at least from studies with skin and prostate cancer cells, suggests that EGCG decreases HDAC activity, increases histone 3 (H3) and 4 (H4) acetylation [200], and downregulates HDAC1, HDAC2, and G9a expression, respectively [200]. 

### 3.5. AtreMorine

E-PodoFavalin-15999 (AtreMorine) is a novel biopharmaceutical compound extract from structural components of the *Vicia faba* L. plant through a non-denaturing biotechnological process, specifically ultrapure lyophilization. The utilization of a non-denaturing process maintains the activity of the active compounds in AtreMorine. The E-PodoFavalin-15999 extract serves as the structural base of AtreMorine and is a natural source of L-DOPA, in addition to containing a plethora of other bioactive substances such as vegetal proteins, unsaturated fatty acids, minerals, vitamins, vegetal fiber, starch, vegetal pigments (carotenes), and vegetal sterols (phytosterols) [201]. This plant is a rich source of 3,4-dihydroxyphenyl-l-alanine (L-DOPA), the dopamine precursor commonly used to treat PD [202,203,204,205]. 

Preclinical screening of AtreMorine with the 6-hydroxydopamine cell culture model and the MPTP rodent model of PD [206,207,208] show that AtreMorine is strongly neuroprotective [206]. Clinically, administration of AtreMorine (5 g/day, p.o.) to patients with Parkinsonian disorders causes a decrease in growth hormone, cortisol, and prolactin levels [208]. AtreMorine enhances catecholamine (dopamine, noradrenaline, adrenaline) levels but with no effect on the serotonergic system in these patients. Furthermore, AtreMorine differentially regulates the pituitary hormones (adrenocorticotropic hormone, growth hormone, prolactin, luteinizing hormone, follicle-stimulating hormone) and the peripheral hormones cortisol, testosterone, and estrogen [209]. In blood (plasma) samples from patients with PD, 5mC levels increase one hour in response to a single oral dose (5 g, p.o.) of AtreMorine [210]. AtreMorine may also alleviate non-motor symptoms associated with PD, including changes in gastrointestinal motility [211]. However, the central and peripheral effects of AtreMorine require further clarification, particularly because AtreMorine causes genotype-dependent effects that involve cytochrome P450 family genes that are associated with drug metabolism, as well as neurodegeneration-linked pathogenic genes. These findings clearly show that AtreMorine is a safe bioproduct in PD, with a strong effect on catecholamines, mainly dopamine.

The effects of L-DOPA on DNA methylation have been widely described, and levodopa increases the levels of different methylation-related proteins [202,203,204,205]. Our group recently showed that AtreMorine reverses alterations in DNA methylation that occur in NDDs such as AD and PD, restoring levels similar to controls [210]. In a triple-transgenic mouse model of AD, AtreMorine was neuroprotective and increased global DNA methylation levels and DNMT3a mRNA expression but decreased HDAC3 expression. The regulation of DNA methylation by AtreMorine is independent of dopamine levels and pathways, suggesting that both dopamine-dependent and -independent mechanisms contribute to its neuroprotective properties [211]. There is an association between different pharmacogenetic geno-phenotypes and the effect of AtreMorine on global DNA methylation. This indicates that, to ensure accurate patient therapy, the pharmacogenetic profiles of patients must be considered [211]. 

The efficacy and pharmacodynamic and pharmacokinetic properties of AtreMorine are highly influenced by genetic and pharmacogenetic factors. The presence of ultra-rapid metabolizers (UM), extensive metabolizers (EM), intermediate metabolizers (IM), and poor metabolizers (PM) associated with various CYP variants, as well as the inheritance of the *APOE-4* allele, all have a significant impact on the AtreMorine-induced dopamine response in PD patients. Although 100% of individuals respond positively to AtreMorine treatment, the magnitude of the response is regulated by the pharmacogenetic profile of each patient. For example, CYP2D6-PMs have lower baseline dopamine levels and response to AtreMorine than CYP2D6-EMs or IMs. CYP2D6-UMs have the highest basal dopamine levels and show the strongest response to AtreMorine [212].

The three primary CYP2C19 geno-phenotypes respond similarly to AtreMorine. However, CYP3A4/5-IMs and CYP2C9-IMs are the best responders, and CYP3A4/5-RMs and CYP2C9-PMs are the poorest responders. Nonetheless, all geno-phenotypes respond with a considerable increase in dopamine levels one hour after AtreMorine treatment (5 g, p.o.) [213]. This highlights the importance of understanding the pharmacogenetic profiles of patients to ensure optimal therapeutic efficacy through precision medicine.

### 3.6. Nosustrophine

Nosustrophine is a novel porcine-derived bioproduct manufactured through non-denaturing biotechnological processes, specifically ultrapure lyophilization. These methods enable the preservation of the biologically active components in Nosustrophine. The nutritional composition of Nosustrophine includes glutamate and aspartate; vitamins B2, B3, D3, and E; calcium; iron; magnesium; zinc; norepinephrine/noradrenaline; dopamine; serotonin; BDNF; neurotrophic tyrosine kinase receptor type 3 (NTRK3/TrkC); and neuropeptide Y, among others [214]. Nosustrophine regulates AD-related gene expression and targets the epigenetic machinery by modulating DNA methylation as well as SIRT and HDAC activity and expression. Mass spectrometric analysis of Nosustrophine extracts revealed the presence of adenosyl-homocysteinase (AHCY), an enzyme implicated in DNA methylation, and nicotinamide phosphoribosyl-transferase (NAMPT), which produces the NAD+ precursor, enhancing SIRT1 activity (Figure 5). Furthermore, Nosustrophine induces deacetylation of the SIRT1 substrate histone H3. SIRT1 activation appears to be a key mechanism which similarly drives the neuroprotective properties of resveratrol and promotes survival by attenuating Bax-dependent apoptosis and the induction of proapoptotic transcription factors. Recent findings indicate that Nosustrophine has significant epigenetic effects against AD and is a novel nutraceutical bioproduct with epigenetic properties (epinutraceutical) that may be effective as a primary or early secondary interventional treatment for AD [214].

## 4. Discussion and Conclusions

It is plausible that epigenetic dysregulation is a key contributor to the development of NDDs. Modifications to the epigenetic machinery (miRNA dysregulation, histone posttranslational modifications, chromatin remodeling, and DNA methylation) are pathogenic in most NDDs, including PD, AD, HD, multiple sclerosis (MS), and ALS. Chronic therapy for these disorders raises the risk of drug toxicity and interactions, which negatively impacts the clinical condition of patients with NDDs. Incorporating pharmacoepigenetics into clinical practice is critical for minimizing drug–drug interactions and adverse drug reactions and for maximizing the overall therapeutic goal of improving cognitive function. To this, the pharmacogenetic apparatus of genes that may be involved in drug safety and efficacy is consolidated by mechanistic, metabolic, pathogenic, pleiotropic, and transporter genes whose expression is regulated by the epigenetic machinery.

Proper nutrition and dietary supplements that enhance brain health (e.g., those containing antioxidant, anti-inflammatory properties) are critical components of effective clinical treatment of patients with age-related cognitive impairment, particularly in individuals with dementia, PD, and other NDDs [215]. Nutritional interventions with biobased natural products have been shown to delay cognitive deterioration in animal models and in patients with NDDs. Epidrugs offer enormous potential for treating NDDs. The use of epinutraceutical bioproducts to treat these diseases is attractive because several natural-derived bioproducts have been produced using nondenaturing biotechnological procedures, which preserve the biological properties of the active ingredients. Natural substances are important sources of bioactive molecules that include epigenetic regulators. Unlike synthetic molecules, most natural molecules are extremely complex and influence more than one epigenetic target (e.g., HDACs, DNMTs, and miRNAs). The concept of “multi-target epidrugs” has emerged in recent years, suggesting that the combination of different epidrugs may enhance therapeutic outcomes. Epinutraceutical bioproducts such as curcumin, AtreMorine, and Nosustrophine are examples of such multi-target epidrugs, as they act on multiple, different epigenetic mechanisms and show beneficial effects in NDDs.

Two significant disadvantages of using chemical agents as epidrugs are their high levels of toxicity and adverse side effects. Biobased natural products have fewer or no adverse side effects, and different natural compounds such as those included in this review show beneficial epigenetic effects in treating NDDs, as supported by data from clinical trials on patients with NDDs.

## Figures and Tables

**Figure 1 pharmaceuticals-16-00216-f001:**
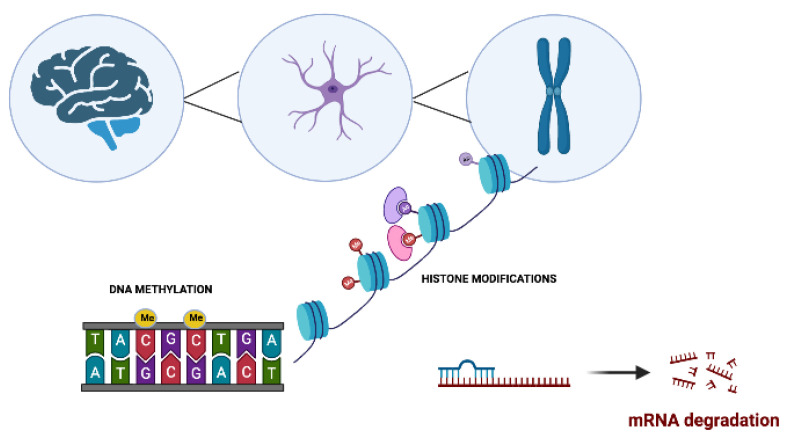
The main molecular epigenetic mechanisms in neurodegenerative disorders. These modifications include DNA methylation, histone modifications, and noncoding RNA-mediated alterations such as microRNA (miRNA) regulation.

**Figure 2 pharmaceuticals-16-00216-f002:**
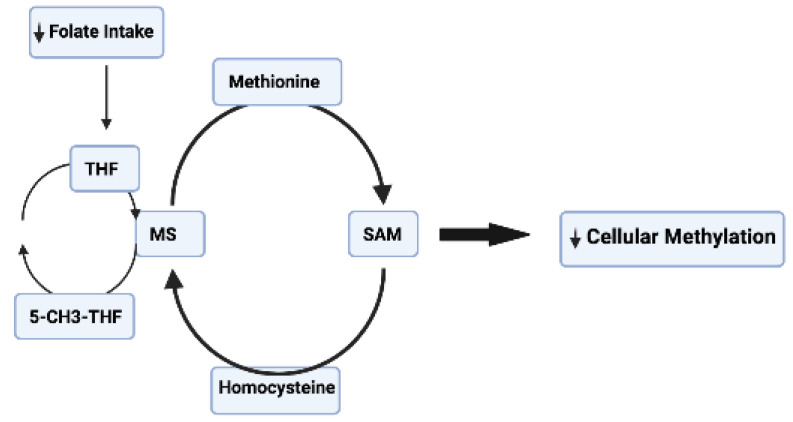
Folate intake regulates the DNA methylation cycle. 5-CH3-THF—5-methyltetrahydrofolate; MS—methionine synthase; SAM—S-adenosyl methionine; THF—tetrahydrofolate.

**Figure 3 pharmaceuticals-16-00216-f003:**
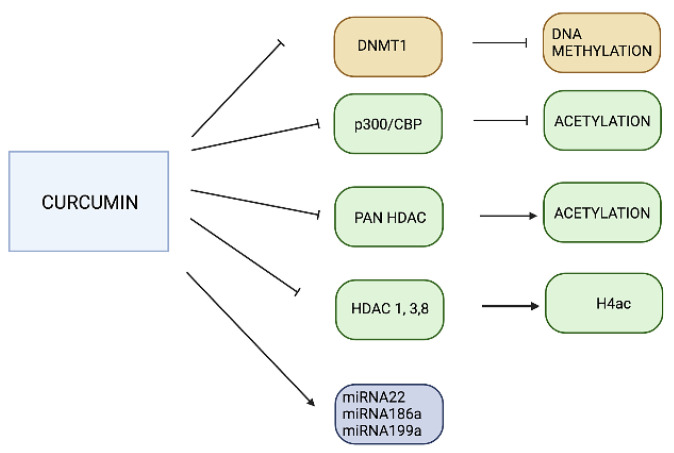
Epigenetic effects of Curcumin. Treatment with curcumin regulates DNA methylation, histone modifications, and miRNA expression in NDDs. NDDs—neurodegenerative disorders.

**Figure 4 pharmaceuticals-16-00216-f004:**
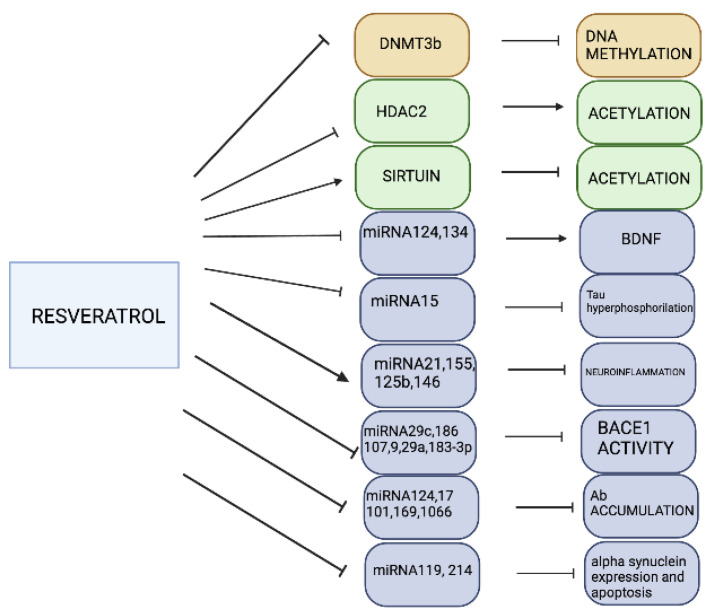
Epigenetic effects of resveratrol. Treatment with resveratrol regulates DNA methylation, histone modifications (acetylation, methylation, phosphorylation, and ubiquitylation), and miRNA expression in NDDs. NDDs—neurodegenerative disorders.

**Figure 5 pharmaceuticals-16-00216-f005:**
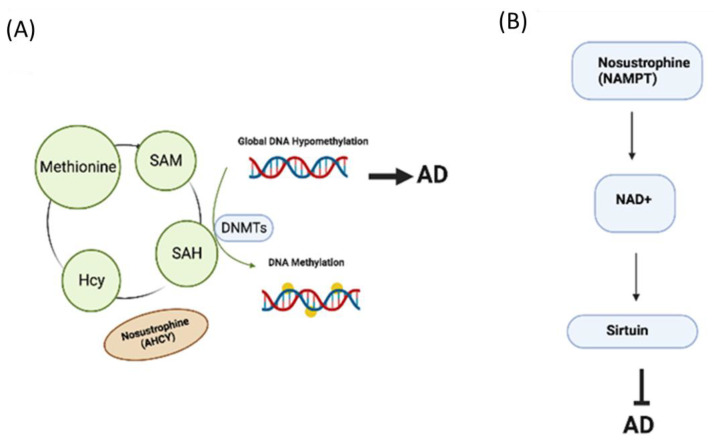
Epigenetic mechanisms of Nosustrophine, a nutraceutical bioproduct with therapeutic potential against AD. (**A**) AHCY increases DNA methylation. (**B**) NAMPT regulates NAD+ production and Sirtuin activity. AD—Alzheimer’s disease; AHCY—adenosylhomocysteinase; DNMT—DNA methyltransferase; Hcy—homocysteine; NAD+—nicotinamide adenine dinucleotide; NAMPT—nicotinamide phosphoribosyltransferase; SAH—S-adenosylhomocysteine; SAM—S-adenosylmethionine.

**Table 1 pharmaceuticals-16-00216-t001:** Nutraceutical compounds implicated in the epigenetic regulation of neurodegenerative diseases.

	DNA Methylation	Histone Modifications	miRNA Regulation
Vitamins A, C, and E	YES	YES	NO
Vitamin B12	YES	NO	NO
Curcumin	YES	YES	YES
Resveratrol	YES	YES	YES
AtreMorine	YES	NO	NO
Nosustrophine	YES	YES	NO

## Data Availability

Data is contained within the article.

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
