# Peer review of "Natural Bioactive Products as Epigenetic Modulators for Treating Neurodegenerative Disorders"

_pharmaceuticals, 2023, doi:10.3390/ph16020216_

Round 1
Reviewer 1 Report
This manuscript provides an overview of the use of natural bioactive products as epigenetic modulators for treating neurodegenerative disorders (NDDs). The author discusses the role of epigenetics in NDDs, specifically focusing on DNA methylation, histone modifications, and miRNA regulation. The manuscript then goes on to review several natural products (curcumin, resveratrol, EGCG, and AtreMorine) that have been studied for their potential use as epidrugs for NDDs. Overall, the manuscript presents a thorough overview of the current state of research on natural bioactive products as epigenetic modulators for NDDs, and the author makes a convincing argument for the potential benefits of these products as treatments. However, there are a few areas that could be improved:
1. The manuscript would benefit from a more detailed discussion of the potential pharmacokinetic and pharmacodynamic properties of the natural products discussed. For example, information on absorption, distribution, metabolism, and excretion (ADME) properties, as well as information on the pharmacological activity of the natural products in the body, would be valuable for readers interested in the pharmaceutical aspect of the research.
2. Include more information on the formulation strategies used to deliver the natural products, and how these strategies may impact the bioavailability of the active compounds. This would be particularly useful for readers interested in the pharmaceutical development of these natural products.
3. The manuscript could benefit from a more in-depth discussion of the safety and toxicity profiles of the natural products discussed. This information would be important for readers interested in the pharmaceutical development of these natural products as well as for future clinical trials.
4. Include more information about the clinical trials that have been done on these natural products, including results, dosage and duration used, population size and demographics, design of the study, etc.
5. The manuscript could benefit from additional data from in vivo and in vitro studies, to support the findings of the study.
6. It could benefit from a more detailed discussion of the potential mechanisms of action of the natural products discussed, and how they may be impacting epigenetic regulation in NDDs.
7. It would be more informative if the author could include more information about the long-term safety and efficacy of these natural products.
8. Provide a more detailed discussion of the potential interactions of the natural products discussed with other drugs and how they may impact the efficacy and safety of these natural products.
9. Include more information on the cost-benefit analysis of these natural products as epidrugs for NDDs, including the cost of production, cost of treatment, and potential savings to the healthcare system.
10. Provide more information on the regulatory aspects of natural products as epidrugs for NDDs, including information on the approval process, patenting, and intellectual property rights.
Author Response
We have incorporated changes that reflect the suggestions provided by the reviewers and have highlighted those changes within the manuscript. We have rephrased the sentences in yellow text. Sentences in blue text are new, and have been included according to the recommendations of the reviewers. The points raised by the reviewers have been addressed as follows (reviewers’ comments are italicized):
REVIEWER 1
Comment 1:
The manuscript would benefit from a more detailed discussion of the potential pharmacokinetic and pharmacodynamic properties of the natural products discussed. For example, information on absorption, distribution, metabolism, and excretion (ADME) properties, as well as information on the pharmacological activity of the natural products in the body, would be valuable for readers interested in the pharmaceutical aspect of the research.
Response:
Thank you to the reviewer. We have included detailed descriptions in the revised manuscript of pharmacokinetic and pharmacodynamic studies for curcumin, resveratrol, epigallocatechin-3-gallate, and AtreMorine. However, Nosustrophine is a relatively new bioproduct and there are currently no published pharmacokinetic studies available on this compound.
Comment 2:
Include more information on the formulation strategies used to deliver the natural products, and how these strategies may impact the bioavailability of the active compounds. This would be particularly useful for readers interested in the pharmaceutical development of these natural products.
Response:
We have provided more detailed information regarding the production process of AtreMorine and Nosustrophine. We have also added new information on the bioavailability of curcumin, resveratrol, and epigallocatechin-3-gallate, and based on the data available, formulation strategies for curcumin and epigallocatechin-3-gallate.
Comment 3:
The manuscript could benefit from a more in-depth discussion of the safety and toxicity profiles of the natural products discussed. This information would be important for readers interested in the pharmaceutical development of these natural products as well as for future clinical trials.
Response:
Thank you to the reviewer. In section 3 of the revised manuscript, we have included detailed information concerning the safety and toxicity profiles of vitamins, curcumin, resveratrol, and epigallocatechin-3-gallate. Safety and toxicity data for Nosustrophine, however, is unavailable.
Comment 4:
Include more information about the clinical trials that have been done on these natural products, including results, dosage and duration used, population size and demographics, design of the study, etc.
Response:
Unfortunately, there are currently few clinical trials investigating the epigenetic effects of natural bioproducts. However, we have included in the revised manuscript those clinical trials that we were able to find (clinicaltrials.gov), specifically those pertaining to curcumin, resveratrol, and epigallocatechin-3-gallate.
Comment 5:
The manuscript could benefit from additional data from in vivo and in vitro studies, to support the findings of the study.
Response:
Thank you to the reviewer. For each of the natural products, we have included data from in vitro and in vivo studies.
Comment 6:
It could benefit from a more detailed discussion of the potential mechanisms of action of the natural products discussed, and how they may be impacting epigenetic regulation in NDDs.
Response:
The aim of our manuscript was to highlight the key epigenetic mechanisms through which plant- and animal-derived bioactive compounds act against neurodegenerative processes. We have not discussed the potential mechanisms of action of natural compounds, but rather focused on the primary objective of our review article, which was to only discuss epigenetic-based mechanisms of action.
Comment 7
It would be more informative if the author could include more information about the long-term safety and efficacy of these natural products.
Response:
Thank you to the reviewer. In section 3 of the revised manuscript, we have included information concerning the long-term safety and efficacy of vitamins, curcumin and epigallocatechin-3-gallate. For more than a decade, AtreMorine has been administered to patients with Parkinson's disease with no reports of adverse effects.
Comment 8
Provide a more detailed discussion of the potential interactions of the natural products discussed with other drugs and how they may impact the efficacy and safety of these natural products.
Response:
We appreciate this comment by the reviewer. Unfortunately, we could not find any studies on interactions of these natural products with other NDD-related drugs.
Comment 9
Include more information on the cost-benefit analysis of these natural products as epidrugs for NDDs, including the cost of production, cost of treatment, and potential savings to the healthcare system.
Response:
Thank you to the reviewer. However, the authors would like to respectfully note that information on the cost-benefit analysis of these natural products as epidrugs for treating neurodegenerative disorders falls outside of the scope and focus of our review article, which is on epigenetics mechanisms of action of natural compounds in neurodegenerative disorders. We have therefore omitted this information from our review article.
Comment 10
Provide more information on the regulatory aspects of natural products as epidrugs for NDDs, including information on the approval process, patenting, and intellectual property rights.
Response:
Thank you to the reviewer. Similar to our response to comment 9, we would like to respectfully point out that information on the regulatory aspects of natural products as epidrugs for neurodegenerative disorders falls beyond the emphasis and scope of our review article. Our main goal was to conduct a literature review, but without delving into regulatory or intellectual property issues, and to provide a critical analysis of the current state of knowledge regarding the efficacy and safety of natural products as potential epitherapeutics for neurodegenerative disorders.
Reviewer 2 Report
In this concise review, the authors briefly summarize current knowledge of the pathological molecular mechanisms of Alzheimer’s and Parkinson’s diseases, and the epigenetic modifications associated with these neurodegenerative diseases. Next, they focus on how some nutraceuticals can improve epigenetic defects in experimental models of these diseases. They point to the beneficial effects of vitamin B12 and other vitamins, polyphenols, AtreMorine, etc… and support that natural bioactive products will offer a complement therapy for neurodegenerative diseases. This review is well written and presented, with clear and informative illustrations.
I just have very minor corrections to make:
Line 110: SNCA gene is only pathogenic in its mutated forms. Please remove “pathogenic” from the sentence.
Line 521: correct Alzheimer spelling.
Author Response
We have incorporated changes that reflect the suggestions provided by the reviewers and have highlighted those changes within the manuscript. We have rephrased the sentences in yellow text. Sentences in blue text are new, and have been included according to the recommendations of the reviewers. The points raised by the reviewers have been addressed as follows (reviewers’ comments are italicized):
Comment 1:
Line 110: SNCA gene is only pathogenic in its mutated forms. Please remove “pathogenic” from the sentence.
Response:
Thank you to the reviewer for pointing this out. The authors apologize for this error, and have made the requested correction.
Comment 2:
Line 521: correct Alzheimer spelling.
Response:
Thank you to the reviewer. The authors apologize for this error and have corrected the spelling.
We sincerely hope that we have addressed the comments to the satisfaction of the reviewers.
Sincerely,
Dr Olaia Martínez-Iglesias
Department of Medical Epigenetics
EuroEspes Biomedical Research Center
Bergondo, 15165
Corunna, Spain
E-mail: epigenetica@euroespes.com